

# Differential lncRNA/mRNA expression profiling and ceRNA network analyses in amniotic fluid from foetuses with ventricular septal defects

Huaming Wang[1], Xi Lin[2], Zecheng Wang[1], Shaozheng He[1], Bingtian Dong[1] and Guorong Lyu[1,3]

[1] Department of Ultrasound, The Second Affiliated Hospital of Fujian Medical University, Quanzhou, Fujian, China
[2] Department of Diagnostic Radiology, Fujian Cancer Hospital of Fujian Medical University, Fuzhou, Fujian, China
[3] Collaborative Innovation Center of Maternal and Child Health Service Technology, Quanzhou Medical College, Quanzhou, Fujian, China

Corresponding author
Guorong Lyu, lgr_feus@sina.com

## ABSTRACT

**Background:** Long noncoding RNAs (lncRNAs) have been shown to be involved in the regulation of numerous biological processes in embryonic development. We aimed to explore lncRNA expression profiles in ventricular septal defects (VSDs) and reveal their potential roles in heart development.

**Methods:** Microarray analyses were performed to screen differentially expressed lncRNAs (DE-lncRNAs) and mRNAs (DE-mRNAs) in the amniotic fluid between the VSD group and the control group. Bioinformatics analyses were further used to identify the functional enrichment and signaling pathways of important mRNAs. Then, a coding–noncoding gene coexpression (CNC) network and competitive endogenous RNAs (ceRNA) network were drawn. Finally, qRT−PCR was performed to verify several hub lncRNAs and mRNAs in the network.

**Results:** A total of 710 DE-lncRNAs and 397 DE-mRNAs were identified in the VSD group. GO and KEGG analyses revealed that the DE-mRNAs were enriched in cardiac development-related biological processes and pathways, including cell proliferation, cell apoptosis, and the Sonic Hedgehog signaling pathway. Four VSD related mRNAs was used to construct the CNC network, which included 149 pairs of coexpressing lncRNAs and mRNAs. In addition, a ceRNA network, including 15 lncRNAs, 194 miRNAs, and four mRNAs, was constructed to reveal the potential regulatory relationship between lncRNAs and protein-coding genes. Finally, seven RNAs in the ceRNA network were validated, including IDS, NR2F2, GPC3, LINC00598, GATA3-AS1, PWRN1, and LINC01551.

**Conclusion:** Our study identified some lncRNAs and mRNAs may be potential biomarkers and therapeutic targets for foetuses with VSD, and described the lncRNA-associated ceRNA network in the progression of VSD.

## INTRODUCTION

Ventricular septal defect (VSD) is one of the most prevalent foetal birth defects; it occurs in 4.3 of 1,000 births and accounts for 39% of cases of congenital heart disease (CHD) in infants (*Liu et al., 2019*). With the development of surgical procedures, some VSD infants have excellent long-term survival; however, early intervention is key to reducing complications and improving prognosis (*Jeon et al., 2020*). To formulate better diagnosis and treatment strategies for VSD, identification of the potential inducements that cause abnormal foetal heart development is a prerequisite. According to epidemiological investigations, genetic factors such as chromosomal or Mendelian syndromes account for approximately 15% of all cases of CHD. The pathogenesis of the remaining 85% of "sporadic" CHDs, which are not associated with chromosomal or Mendelian syndromes, has proven intractable to genetic investigation (*Hopkins, Dugoff & Kuller, 2019*). These "sporadic" CHDs are presumed to be the result of multiple genetic and environmental factors, suggesting incomplete penetrance. Epigenetic factors reflect the interactions between genetic and environmental factors, which can be explained by the differences in the genetic buffering capacity between individuals (*Lim, Foo & Chen, 2021*).

Recently, studies have shown that the development of the foetal heart is a complex and dynamic process regulated by the correct spatiotemporal expression of noncoding RNAs (ncRNAs) (*Chahal, Tyagi & Ramialison, 2019*). Long noncoding RNAs (lncRNAs), which are the most abundant isoform of ncRNAs, have been shown to participate in the regulation of tissue or cellular function by epigenetic mechanisms (*Ye et al., 2022*). LncRNAs can activate or repress gene expression in several ways to affect foetal heart development. For example, lncRNA uc.4 inhibits cell differentiation in heart development by altering DNA methylation (*Zhang et al., 2018*). Enhancer lncRNAs are involved in the maintenance of cardiac stem cell pluripotency and cardiomyocyte differentiation (*Devaux et al., 2015*). In addition, lncRNAs can act as competitive endogenous RNAs (ceRNAs) to regulate the expression of downstream target genes by competitively binding with miRNAs. *Han et al. (2016)* found that lncRNA-H19 facilitates Sox6 expression by binding with miR-19b to promote apoptosis and inhibit cell proliferation in P19CL6 cells during late-stage cardiac differentiation. Several studies have shown that abnormal expression of lncRNAs can cause corresponding cardiac defects. *Wang et al. (2018)* found that the expression of lncRNA-HA117 in the myocardial tissue of children with Tetralogy of Fallot (ToF) was increased and negatively correlated with prognosis. *Li et al. (2017a)* found that lncRNA-TUC40 expression was significantly increased in the heart tissues of VSD foetuses. Based on these findings, understanding the network structure and regulatory mechanism of lncRNAs could profoundly impact the diagnosis and treatment of VSD.

Amniotic fluid (AF) is a complex and dynamic biological fluid, that is rich in foetal cell-free DNA and RNA, and contains 100–200 times more foetal nucleic acids than maternal plasma (*Kamath-Rayne et al., 2014*). Since AF is in direct contact with the foetal skin, urinary system, lungs, oropharynx, and gastrointestinal tract, it can directly reflect the growth and development of the foetus (*O'Neill et al., 2018*). Recently, studies have shown that AF provides dynamic ncRNA information about foetal pathophysiological status

inside the uterus. *Vizitiu et al. (2019)* showed that miR-99a was significantly upregulated in the AF of foetuses with trisomy 21 syndrome. *Xie et al. (2017)* found that AF-derived miR-299-5p and miR-300 could be used as biomarkers to evaluate foetal kidney function and prenatally diagnose congenital hydronephrosis. In the clinic, AF is often collected to detect chromosomal abnormalities and pregnancy-related complications. However, the relationship between AF-derived lncRNAs and VSD remains to be fully exploited. Although our previous publication showed that some lncRNAs were differentially expressed in the AF-derived exosomes of VSD fetuses (*Yang et al., 2021*), there are still many VSD-related lncRNAs have not been excavated. In addition, research has shown that some lncRNAs are enriched in exosomes, while others are barely present, indicating that some lncRNAs are selectively sorted into exosomes (*Sun et al., 2018*). Therefore, we considered that lncRNA in AF supernatant can provide a more comprehensive analysis of foetal development.

With the development of high-throughput analysis technology, it is possible to massively analyze the transcriptome in parallel (*Liguori et al., 2018*). Compared with miRNAs whose functions are relatively well studied, lncRNAs are still poorly annotated and contain repetitive elements, making the analysis of RNA-seq data challenging (*Weirick et al., 2016*). Therefore, in this study, a lncRNA microarray was used to screen differentially expressed lncRNAs (DE-lncRNAs) and mRNAs (DE-mRNAs) in AF between the VSD group and the control group. Then, VSD-related lncRNAs and mRNAs were selected by bioinformatics analyses and verified by qPCR. Based on these analyses, we predict the potential role of lncRNAs in VSD and provide new biomarkers for the diagnosis and treatment of VSD.

## MATERIALS AND METHODS

### Patient recruitment and ethics statement

In this study, pregnant women were recruited from the Second Affiliated Hospital of Fujian Medical University. All pregnant women had indications for prenatal diagnosis and were assessed by prenatal ultrasound and echocardiography. For safety and reasons of ethical constraints, amniotic fluid sampling was only performed when pregnant women had indications for prenatal diagnosis and without additional burden for the mother or fetus. Ultimately, 22 singleton pregnant women aged 19–34 years between 20 and 22+6 weeks of gestation were included in the research. Of these, 11 pregnant women who carried a VSD foetus were in the VSD group, and the other 11 pregnant women who carried a structurally normal foetus but had a high risk Down's screening were in the control group. Both groups of pregnant women had no pregnancy complications and poor life histories such as hypertension, diabetes, smoking, or alcoholism. All cases were tested for chromosome anomalies by karyotype. Foetuses with abnormal karyotypes and extracardiac major associated anomalies were excluded. There were no differences in maternal age, the number of pregnancies, gestational age, or body mass index (BMI) between the VSD group and the control group ($P > 0.05$). These clinical characteristics are shown in Tables 1 and S1. This study was performed in accordance with the principles of

Table 1 Clinical characteristics of the VSD group and the control group.

| Characteristics | VSD group ($n$ = 11) | Control group ($n$ = 11) | $P$ value |
|---|---|---|---|
| Maternal age (years) | 30.82 ± 3.9 | 31.64 ± 5.2 | 0.681 |
| Gestational age (weeks) | 21.18 ± 0.9 | 20.64 ± 0.8 | 0.144 |
| Number of deliveries | 1.91 ± 0.8 | 2.09 ± 0.8 | 0.614 |
| Body mass index (kg/m$^2$) | 22.09 ± 1.0 | 21.82 ± 1.3 | 0.273 |

the World Medical Association Declaration of Helsinki and licenced by the Ethics Committee of the Second Affiliated Hospital of Fujian Medical University (NO. 2021-073).

## RNA extraction

AF was extracted by amniocentesis from pregnant women and specimens with gross maternal blood or meconium contamination were discarded. AF samples were centrifuged at 1,600×$g$ for 10 min at 4 °C to remove amniocytes, and then supernatant samples were stored at −80 °C until RNA extraction. Total RNA was extracted from AF supernatant using TRIzol LS Reagent (Invitrogen, Carlsbad, CA, USA) according to the manufacturer's protocol. Then, RNA quantity and quality were measured by a NanoDrop ND-1000.

## RNA labelling and array hybridization

In this study, we selected Arraystar Human LncRNA Microarray V5.0, which is designed for the global profiling of human lncRNAs and protein-coding transcripts and can detect approximately 39,317 lncRNAs and 21,174 coding transcripts. The specific operations were as follows, each RNA sample was amplified and transcribed into fluorescent cRNA along the entire length of the transcripts without 3′ bias utilizing a random priming method (Arraystar Flash RNA Labelling Kit, Arraystar). The labelled cRNAs were purified using an RNeasy Mini Kit (Qiagen). The concentration and specific activity of the labelled cRNAs (pmol Cy3/μg cRNA) were measured by a NanoDrop ND-1000. Total of 1 μg of each labelled cRNA was fragmented by adding 5 μl 10 × Blocking Agent and 1 μl of 25 × Fragmentation Buffer, and then heating the mixture at 60 °C for 30 min. Finally 25 μl 2 × GE Hybridization buffer was added to dilute the labelled cRNA. Total of 50 μl of hybridization solution was dispensed into the gasket slide and assembled onto the lncRNA expression microarray slide. The slides were incubated for 17 h at 65 °C in an Agilent Hybridization Oven. The hybridized arrays were washed, fixed, and scanned using the Agilent DNA Microarray Scanner (part number G2505C). The microarray work was performed by Kang Chen Bio-tech, Shanghai, China.

## Data analysis

Agilent Feature Extraction software (version 11.0.1.1) was used to analyse acquired array images. Then, the GeneSpring GX v12.1 software package (Agilent Technologies) was used to perform quantile normalization and subsequent data processing (*Zahurak et al., 2007*). Differentially expressed lncRNAs (DE-lncRNAs) and mRNAs (DE-mRNAs) between the two groups were identified through Fold Change filtering and $P$ value with the R-based

limma software package (*Diboun et al., 2006*). Finally, heatmap and volcano plots were generated for significantly differentially expressed genes using the "ggplot2 package" of R (*Wickham, 2016*). The threshold used to screen upregulated or downregulated lncRNAs and mRNAs was a | fold-change | ≥ 2.0 ($P < 0.05$). The microarray data discussed in this paper have been deposited in the NCBI Gene Expression Omnibus and are accessible with the GEO Series accession number GSE204935.

## Bioinformatics analysis

To identify the potential role of differentially expressed genes in VSD, Gene Ontology (GO) annotation and the Kyoto Encyclopedia of Genes and Genomes (KEGG) were used to illuminate the functions of the DE-mRNAs (*The Gene Ontology Consortium, 2017*; *Kanehisa & Goto, 2000*). GO analysis provides the key functional classifications for DE-mRNAs, which include biological process (BP), cellular component (CC), and molecular function (MF). KEGG reveals the relationship networks and disease information for DE-mRNAs. Fisher's exact test was used to classify the GO and KEGG categories. The threshold of significance was $P$ value < 0.05 and FDR < 0.1.

## Construction of the CNC network and ceRNA network

According to the above analysis, we selected core DE-mRNAs closely related to VSD for further study. Based on the expression level of all DE-lncRNAs and core DE-mRNAs, the Pearson correlation coefficient (PCC) was calculated by the "coXpress package" of R (*Watson, 2006*). The DE-lncRNAs and DE-mRNAs with PCC > 0.95 and $P$ value < 0.05 were selected to construct the CNC network.

To further explore the potential role of lncRNAs, a ceRNA (lncRNA–miRNA–mRNA) network based on core DE-mRNAs was constructed. First, the miRcode database (v11) was used to predict miRNA targets on lncRNAs (*Jeggari, Marks & Larsson, 2012*), and the TargetScan database (v7.2) was used to predict miRNA targets on mRNAs (*Agarwal et al., 2015*). The thresholds for the algorithms were conservation scores >50 for miRcode, and target scores ≥70 for TargetScan. Then, the lncRNA–miRNA–mRNA interaction network was constructed using the overlapping miRNAs that harboured both lncRNA and mRNA binding targets. The Cytoscape (v2.8.2) program was used to construct and visualize the CNC network and ceRNA network (*Shannon et al., 2003*).

## Quantitative real-time reverse transcription PCR (qRT–PCR)

Total RNA was extracted from AF supernatant using TRIzol LS Reagent (Invitrogen, Carlsbad, CA, USA) and then cDNA was synthesized from 1 ug of total RNA using PrimeScript™ RT reagent Kit with gDNA Eraser (Takara, Dalian, China). cDNA was amplified and quantified using TB Green® Premix Ex Taq™ II (Takara, Dalian, China). The cycling parameters were as follows: the first step, 95 °C for 30 s; the second step, 95 °C for 5 s and 60 °C for 34 s, for 40 cycles; and the third step, for melting 95 °C for 15 s, 60 °C for 60 s and 95 °C for 15 s. Each sample was measured in triplicate and the sequences of the

primers used are shown in Table S2. The obtained data of lncRNAs and mRNAs expression levels were calculated and normalized by the $2^{-\Delta\Delta Ct}$ method as follows: $\Delta Ct_{(test/control)}$ = average value $Ct_{(target\ gene)}$ − average value $Ct_{(GAPDH)}$; $\Delta\Delta Ct = \Delta Ct_{(test/control)}$ − average value $\Delta Ct_{(control)}$.

## Statistical analysis

Heatmaps and volcano plots based on the expression levels of genes were generated and visualized using R statistical software. Fisher's exact test was employed to evaluate the significance of GO or KEGG enrichment. The expression levels of lncRNAs and mRNAs were calculated *via* an unpaired t-test between the VSD and control groups.

All quantitative data are expressed as the mean ± SD and were analysed using SPSS 21.0 (SPSS Inc., Chicago, IL, USA). $P < 0.05$ was considered to indicate a statistically significant difference (*Mushquash & O'Connor, 2006*).

## RESULTS

### Differentially expressed lncRNAs and mRNAs

To evaluate the differential expression of AF-derived lncRNAs and mRNAs between the VSD and control groups, AF supernatant from five VSD foetuses and five control foetuses was obtained by a random method and subjected to microarray analysis. Box plots were used to compare the intensities and quality of microarray data (Figs. 1A and 1B). The result showed that the distributions of intensities in different samples were similar after normalization. A volcano plot was used to assess the variation in lncRNA and mRNA expression between the two groups (Figs. 1C and 1D). Finally, hierarchical clustering was used to show distinguishable lncRNA and mRNA expression profiles (Figs. 1E and 1F). With the cut-off criteria of a | fold change | ≥ 2 and *P* value < 0.05, 710 lncRNAs were significantly differentially expressed between the two groups, of which 240 were upregulated and 470 were downregulated in the VSD group. Similarly, 397 mRNAs were significantly differentially expressed between the two groups, of which 106 were upregulated and 291 were downregulated in the VSD group. The expression level of the up- and down-regulated DE-mRNAs and DE-lncRNAs was showed in Tables S3 and S4.

### Characteristics of differentially expressed lncRNAs and mRNAs

To further analyze the DE-lncRNAs and DE-mRNAs, we summarized some general characteristics of these genes, such as type distribution, genomic location distribution, and length. As shown in Figs. 2 A and 2B, in both up- and down-regulated lncRNAs, intergenic lncRNAs are the largest types, and intron sense overlapping lncRNAs are the least type. The results of positioning showed that up- and down-regulated lncRNAs were mainly distributed on chromosomes 1 and 2, respectively, while both the up- and down-regulated mRNAs were mainly distributed on chromosomes 1 (Figs. 2C, 2D and S1). The length distribution showed that most DE-lncRNAs were distributed within 0.5–1 kb; however, the DE-mRNAs were mainly distributed at 2–4 kb in length. These results prompt that chromosome 1 and 2 may contain more genes related to the VSD process, and compared

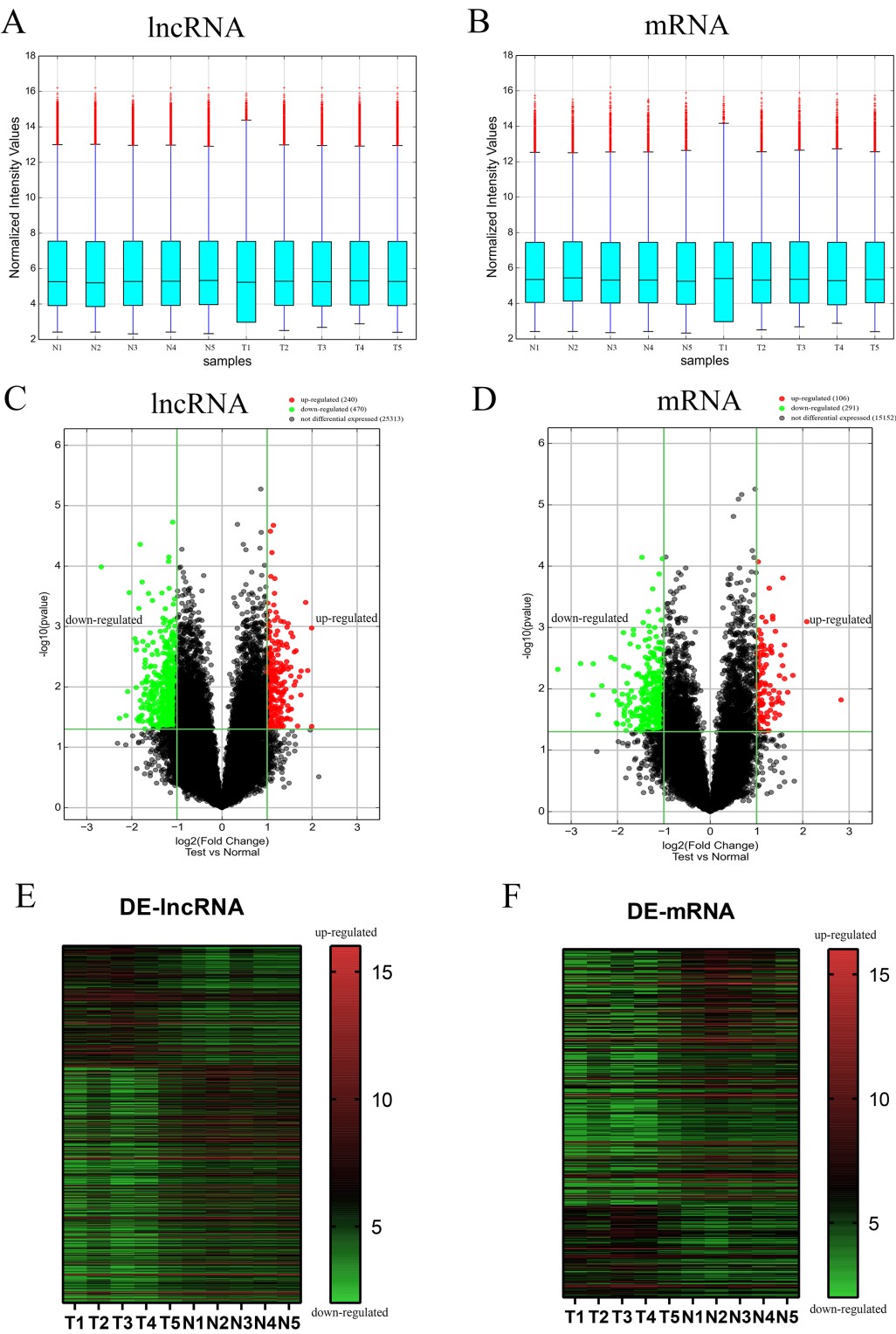

**Figure 1 Different expression profiles of lncRNAs and mRNAs between VSD and control samples.**
(A, B) Box plots of lncRNAs and mRNAs showing the distributions of intensities from all samples.
The distributions were similar after normalization. (C, D) Volcano map of lncRNAs and mRNAs.
The horizontal green line represents $P = 0.05$, and the vertical green lines correspond to 2.0-fold up and
down. Data points in black indicates genes with no significant difference, red indicates upregulated genes,

**Figure 1** (continued)
and green indicates downregulated genes. (E, F) Heatmap of lncRNAs and mRNAs. The abscissa represents different samples, and the ordinate represents different genes. The red boxes indicate upregulated genes, and the green boxes indicate downregulated genes. Genes expressed at the average level are in black. "T" stands for VSD sample, "N" stands for control sample.

with other types of lncRNA, intergenic lncRNAs are more critical in the formation of cardiac defects.

## GO and pathway analysis

To explore the mechanisms and pathways during the progression of VSD, the biological meaning and systematic features of the significant DE-mRNAs were first annotated by GO ontology. A total of 713 GO terms were significantly enriched. The top ten terms of the GO ontology results are shown in Fig. 3A. BP analysis indicated that the DE-mRNAs were enriched in heart development, cell adhesion, and regulation of cell proliferation. CC analysis indicated that the DE-mRNAs were enriched in cell junction, cell-substrate junction, and extracellular matrix. MF analysis indicated that the DE-mRNAs were enriched in kinase regulator activity and protein-containing complex binding.

KEGG pathway analysis was used to investigate the wire diagrams of molecular interactions, reactions, and relations related to the DE-mRNAs. The top ten enriched KEGG pathways are shown in Fig. 3B including the Sonic Hedgehog signaling pathway (Shh), the Phosphatidylinositol signaling system, apoptosis, and the regulation of the actin cytoskeleton. In addition, KEGG disease enrichment analysis showed that DE-mRNAs were involved in congenital malformations, cardiovascular diseases, and nervous system diseases (Fig. 3C). Of these, the most enriched disease was congenital malformations with 24 genes annotated with this term. These results show that DE-mRNAs are rich in biological activities and signal pathways related to heart development, including cell adhesion, cell proliferation and apoptosis, tissue development, and regulation of Shh signaling pathway.

## Construction of the CNC network and ceRNA network

According to GO and KEGG analysis, four core mRNAs, including IDS, ID1, NR2F2, and GPC3, were selected to construct a CNC network because they are closely connected to cardiac biology. Among the coexpression analysis, four core mRNAs and 141 lncRNAs comprise the CNC network, which includes 149 pairs of coexpressing lncRNAs and mRNAs. As shown in Fig. 4A, a core mRNA could have a correlation with many lncRNAs, and most of these relationship pairs presented a positive correlation. To our knowledge, ceRNAs play important roles in the cis-regulation of lncRNAs. Therefore, we constructed a ceRNA network to reveal the regulatory mechanism between lncRNAs and mRNAs in the CNC network. In the ceRNA network, the miRcode and TargetScan databases were used to predict miRNA targets on lncRNAs and mRNAs, respectively. A total of 194 overlapping miRNAs were selected to construct the ceRNA network. To better show the specific ceRNA network, we constructed the network of up-regulated and down-regulated genes, respectively. Eventually, the down-regulated ceRNA network includes 11 lncRNAs, 129

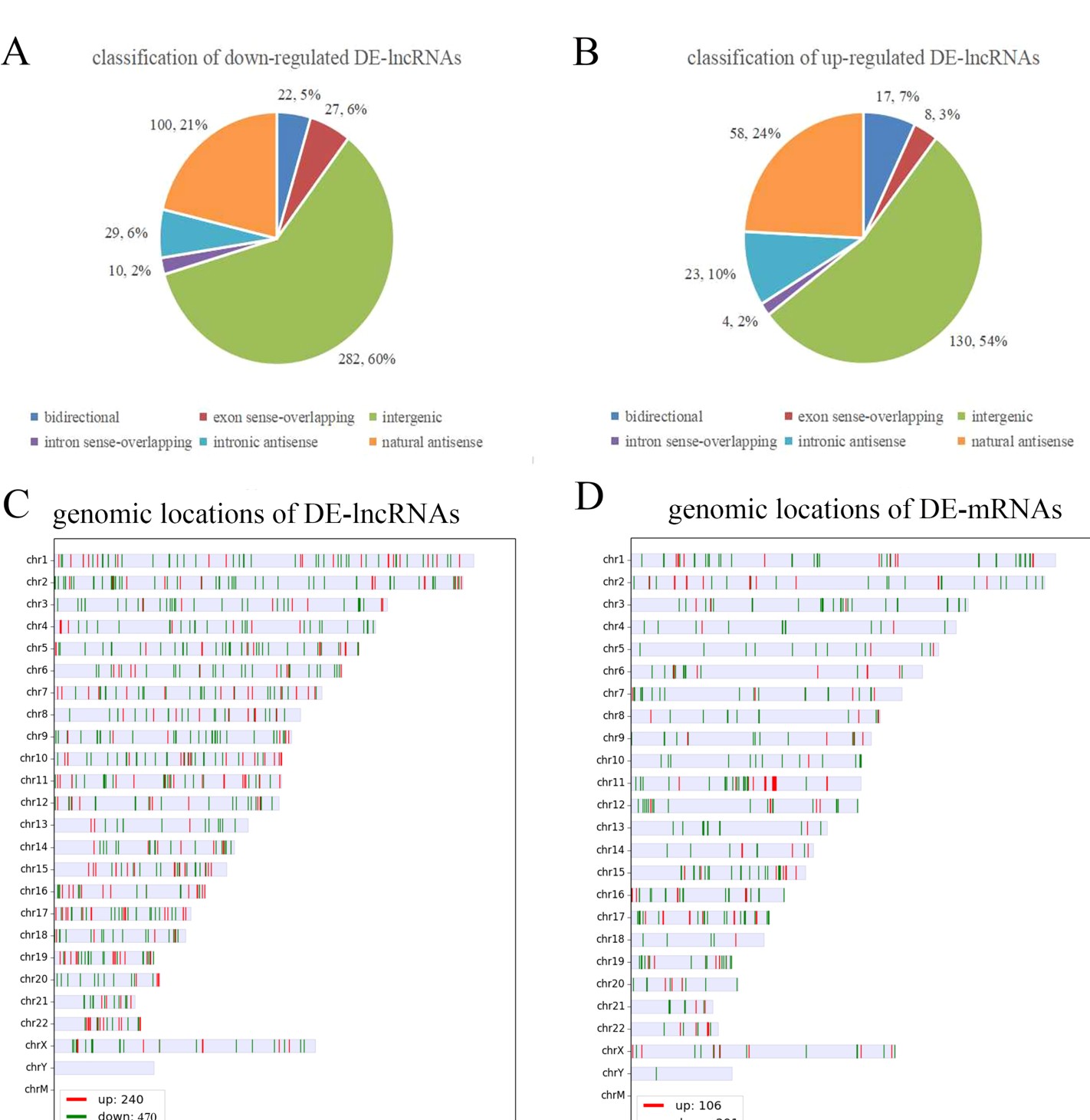

**Figure 2 Characteristics of the differentially expressed lncRNAs and mRNAs.** (A, B) Type distribution of the down- and up-regulated lncRNAs. (C, D) Distribution of lncRNA and mRNA genomic locations. Red indicates upregulated genes, and green indicates downregulated genes.

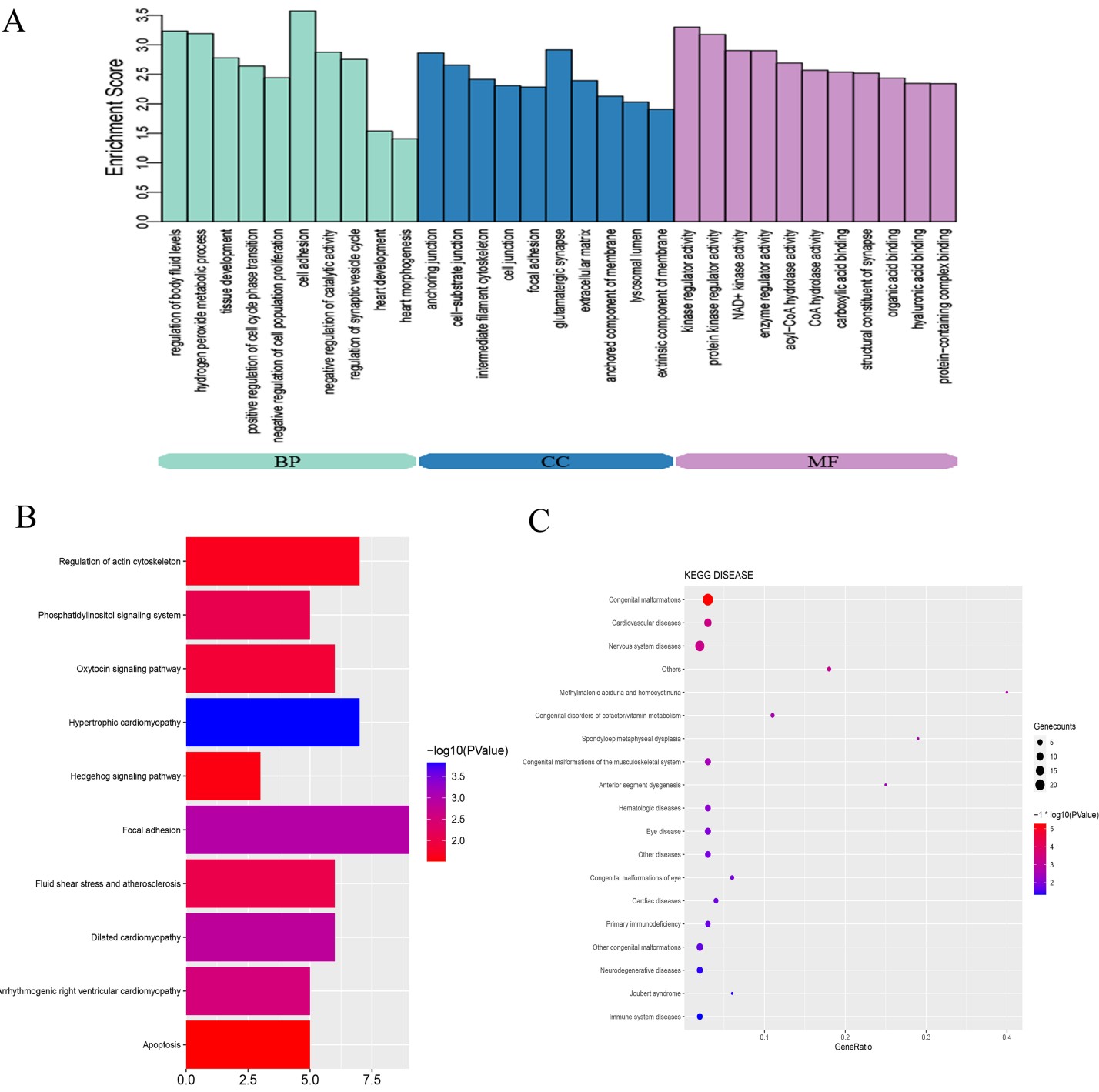

**Figure 3 Bioinformatics analysis of DE-mRNAs.** (A) Top ten GO enriched terms for the DE-mRNAs between the VSD and control groups. The abscissa represents the GO term name, and the ordinate represents the enrichment score. (B) Top ten enriched KEGG pathways of DE-mRNAs between the VSD and control groups. The abscissa represents the enrichment score, and the ordinate represents the KEGG pathway name. (C) KEGG disease enrichment analysis of DE-mRNAs. The abscissa represents the gene ratio, and the ordinate represents the KEGG disease name. The size and colour of the bubbles represent the number of genes enriched in the disease and the value of −log10 ($P$ value), respectively. GO, Gene Ontology; BP, Biological Process; CC, Cellular Component; MF, Molecular Function; KEGG, Kyoto Encyclopedia of Genes and Genomes.

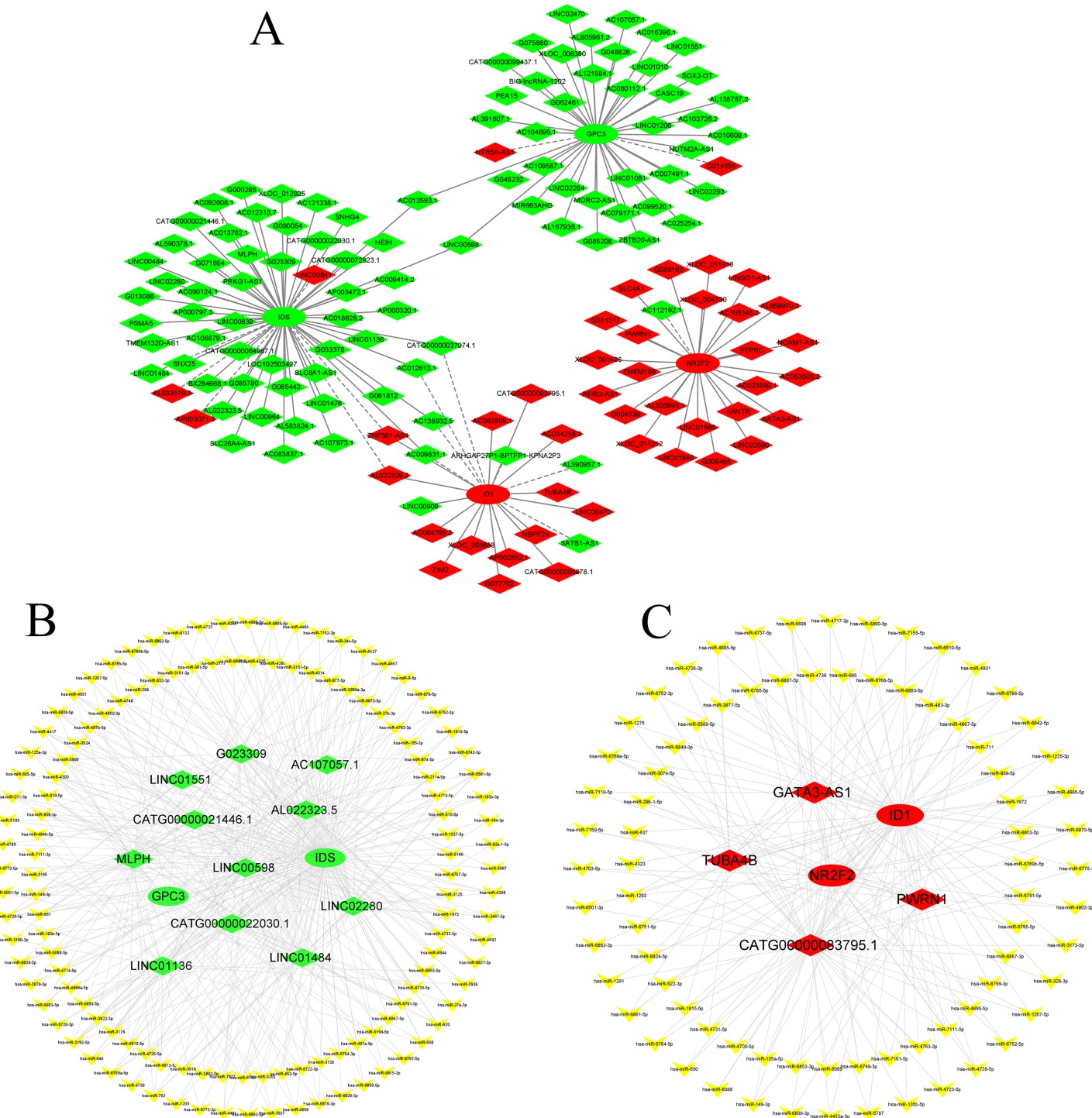

**Figure 4 CNC network and ceRNA network.** (A) CNC network. The ellipse represents mRNAs and the diamond represents lncRNAs. Red and green represent up- and downregulated RNAs, respectively. The solid line indicates a positive correlation, and the dashed line indicates a negative correlation. (B, C) ceRNA network of downregulated and upregulated genes. The ellipse represents mRNAs, the diamond represents lncRNAs, and the V shape represents miRNAs. Red and green represent up- and downregulated RNAs, respectively.

**Table 2 qPCR detection of differential expression of lncRNAs and mRNAs in the VSD group and the control group.**

| RNA | VSD group (n = 11) | | Control group (n = 11) | | Fold chang | P value |
|---|---|---|---|---|---|---|
| | Mean | SD | Mean | SD | | |
| IDS | 0.42 | 0.22 | 1.05 | 0.34 | 0.40 | 0.000 |
| ID1 | 1.93 | 1.25 | 1.26 | 0.93 | 1.53 | 0.164 |
| NR2F2 | 3.89 | 1.63 | 1.45 | 1.62 | 2.68 | 0.002 |
| GPC3 | 0.47 | 0.29 | 1.14 | 0.66 | 0.41 | 0.006 |
| AL022323.5 | 1.20 | 1.51 | 2.99 | 4.16 | 0.40 | 0.193 |
| LINC00598 | 0.44 | 0.34 | 1.36 | 1.09 | 0.32 | 0.015 |
| GATA3-AS1 | 3.88 | 2.06 | 1.43 | 1.04 | 2.71 | 0.002 |
| PWRN1 | 5.60 | 2.97 | 1.11 | 0.56 | 5.05 | 0.000 |
| LINC01551 | 0.34 | 0.14 | 1.02 | 0.18 | 0.33 | 0.000 |

miRNAs, and two mRNAs, while, the up-regulated ceRNA network includes four lncRNAs, 79 miRNAs, and two mRNAs. (Figs. 4B and 4C).

## Validation of RNA expression by qPCR

According to the GO/KEGG analysis and CNC/ceRNA network, we examined the expression of four core mRNAs (IDS, ID1, NR2F2, and GPC3) and five lncRNAs (AL022323.5, LINC00598, GATA3-AS1, PWRN1, and LINC01551) to confirm the reliability of our microarray results. To increase the accuracy of the results, qPCR was performed in five VSD and five healthy samples used for microarray, and in six additional VSD and six additional healthy samples. The qPCR results showed that seven RNAs (IDS, NR2F2, GPC3, LINC00598, GATA3-AS1, PWRN1, and LINC01551) were consistent with the results of microarray analysis, while two RNAs (ID1, and AL022323.5) were not significantly different between the two groups (Table 2 and Fig. 5).

## DISCUSSION

The development of the heart is a complex and dynamic process regulated by a variety of ncRNAs and mRNAs. Although transcriptomic analyses of CHD have been reported in several studies, information about lncRNA expression and the mechanism by which lncRNAs contribute to the development of VSD is limited. In the present study, we compared the expression profiles of the lncRNAs and mRNAs between the VSD group and the control group using the microarray technique. We identified 710 dysregulated lncRNAs and 397 dysregulated mRNAs and summarized their general characteristics. Through bioinformatics analysis, we found that these dysregulated genes were enriched in cardiac development-related biological processes and pathways. Subsequently, a CNC network and ceRNA network were constructed to predict the mechanisms and functional roles of gene regulation. The qPCR results confirmed that NR2F2, GATA3-AS1, and PWRN1 were upregulated, and IDS, GPC3, LINC00598, and LINC01551 were downregulated in VSD.

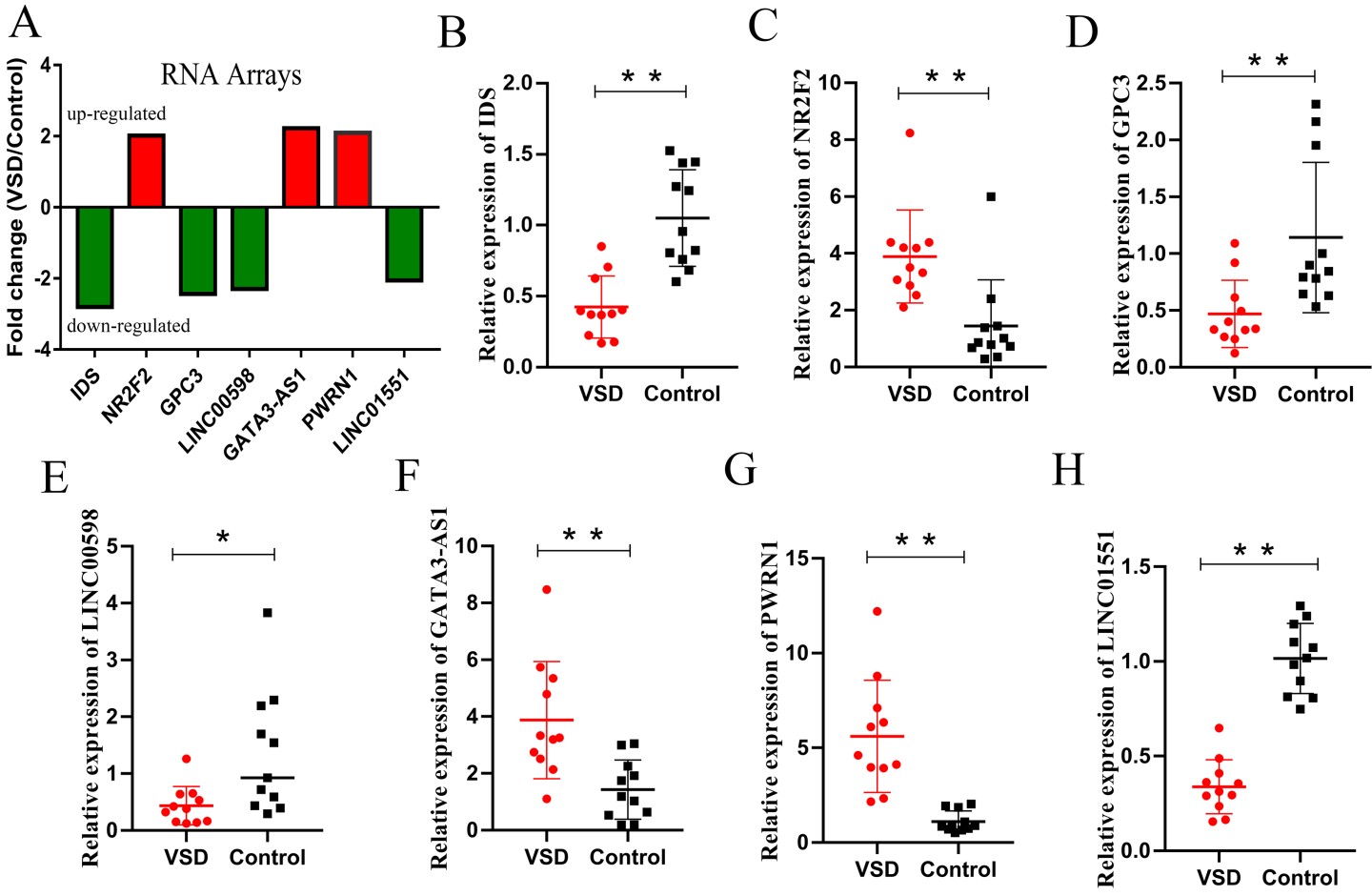

**Figure 5 Relative expression levels of candidate lncRNAs and mRNAs from the RNA array and qPCR.** (A) Fold change of seven candidate RNAs from the RNA array. Red represents upregulated genes, while green represents downregulated genes. (B–H) Expression levels of seven candidate RNAs from qPCR. An asterisk (*) indicates $P < 0.05$ and two asterisks (**) indicate $P < 0.01$.

To understand the role of lncRNAs in the pathogenesis of VSD, we have generated an integrated analysis of dysregulated lncRNAs and mRNAs. GO and KEGG analyses showed that DE-mRNAs were enriched in cell adhesion, cell proliferation and apoptosis, tissue development, and regulation of Shh signaling pathway. The overall growth of the heart during embryogenesis involves multiple overlapping stages of cell proliferation, apoptosis, and morphogenesis (*Yu et al., 2019*). The decreased proliferation or increased apoptosis of cardiomyocyte will lead to cardiac defects, including developmental defects in heart valves and double outlet right ventricle (*Miao et al., 2020*; *Zhang et al., 2019*). In addition, cell adhesion is essential for maintaining the integrity of the chambers and compartments of the heart (*Li et al., 2017b*). Shh is a key signaling pathway during cardiac cell regeneration and heart ontogenesis. Abolishing the Shh signaling pathway for a brief period during embryogenesis is sufficient to bring about irreversible cardiac deformities (*Costa et al., 2017*). The above studies have showed that the biological activities of DE-mRNAs are closely related to cardiac biology and may be the molecular mechanism of VSD development.

Through biological information analysis and literature review, four mRNA, including IDS, ID1, NR2F2, and GPC3, were selected for the next study. These mRNAs are related to cardiac development. For instance, IDS contributes to aberrant heart development and cardiomyocyte differentiation disorder by affecting the catabolism of GAGs (*Matsuhisa & Imaizumi, 2021*). ID1 encodes a transcription regulator that is involved the differentiation of cardiac precursors by regulating apoptosis and proliferation of cardiomyocyte (*Hu et al., 2019*). Pathogenic mutations in the NR2F2 and GPC3 will lead to many cardiac defects, such as VSD, Tetralogy of Fallot and double outlet right ventricle (*Duong et al., 2018*; *Capurro et al., 2008*). Interestingly, these four genes are related to the Shh signaling pathway. ID1 can activate the Shh signaling pathway to maintain cell stemness (*Sun et al., 2019*). Silencing IDS and NR2F2 can significantly inhibit the expression of Shh downstream target genes (*Costa et al., 2017*; *Li et al., 2013*). The loss of GPC3 function selectively reduces Shh signaling in cardiomyocyte (*Liu, Wierbowski & Salic, 2022*). Therefore, we consider that these DE-mRNAs may affect the biological activity of cardiomyocyte through the Shh signaling pathway, leading to the formation of CHD.

How lncRNAs regulate molecular and cellular functions remains largely elusive. One possibility is that lncRNAs regulate the expression levels of coding RNAs and add intricacy to the regulatory network of cardiac gene expression (*Ye et al., 2022*; *Kim et al., 2021*). Therefore, we selected lncRNAs related to the above mRNAs for further analysis. First, according to the ceRNA hypothesis, we constructed a CNC network and ceRNA network to reveal the potential regulatory relationship between lncRNAs and protein-coding genes. Second, we selected conserved lncRNAs in the network for qPCR validation. Finally, we identified four lncRNAs that were differentially expressed between the VSD group and the normal group, including LINC00598, GATA3-AS1, PWRN1, and LINC01551. Although the mechanism of these four lncRNAs in CHD is not clear, many research have shown that they are related to the cell cycle, cell proliferation and apoptosis. For instance, knockdown of LINC00598 can inhibit cell proliferation by inducing G0/G1 cell cycle stagnation (*Jeong et al., 2016*). Knockdown of GATA3-AS1 and LINC01551 markedly limited cell invasion abilities, cell proliferation, and cell viability (*Liu, Xu & Li, 2021*; *Xue & Cao, 2020*). PWRN1 overexpression suppress xenograft growth *in vivo* and cancer cell migration *in vitro* (*Jiang, Wang & Lu, 2020*). Recently, numerous studies have revealed that lncRNA can indirectly regulate mRNA expression levels by competitively binding to a miRNA response element (*Salmena et al., 2011*). Moreover, published reports have confirmed that lncRNA can lead to CHD through this regulation mechanism. For example, FGD5-AS1 is an hub lncRNA in the TOF heart ceRNA network and can regulate SMAD4 by binding with hsa-miR-421 (*Zhang et al., 2021*). It is worth mentioning that the four verified lncRNAs as a miRNA sponge are widely reported in cancers, such as the GATA3-AS1/miR-495-3p/CENPU axis in breast cancer (*Lin et al., 2021*), and the PWRN1/miR-214-5p axis in osteosarcoma (*Shi et al., 2020*). Based on the above analysis, we speculated that lncRNAs with ceRNA activity may be regulate the cell cycle, cell proliferation, and apoptosis by regulating heart-related mRNA and lead to the development of the CHD.

The genes mentioned above seem to be critical for the occurrence of CHD, and gene regulation mediated by ncRNAs may provide new strategies for the treatment of CHD. In the past few years, several ncRNA-based drugs have been in phase II or III clinical development. Most of these drugs regulate the expression of downstream genes by mimicking or antagonizing the function of ncRNA to achieve therapeutic effects (*Pei, Zhou & Liu, 2022*). For example, an intravenous injection of a miR-378 mimic prevents excessive myocardial fibrosis in a mouse model of transverse aortic constriction (*Yuan et al., 2018*). Although some potential therapeutic targets were found in our study, the clinical application of ncRNA still faces key challenges, including tolerability, delivery, specificity, and potential side effects (*Winkle et al., 2021*). Therefore, a detailed understanding of the biological and molecular mechanisms underlying ncRNA-mediated CHD development is key to translating such knowledge into clinical usage.

There are several limitations of this study that should be discussed. First, a small sample size is the main limitation of this study. In addition, the assessment of the correlation between lncRNAs and mRNAs was based on a comprehensive analysis of five pairs of samples, which may have inevitable biases. Therefore, we need to increase the sample size for future research. Second, the exact influence mechanism of mRNAs and lncRNAs validated by qRT–PCR on VSD has not been investigated in depth. The molecular functions and molecular mechanisms mediated by lncRNAs still need to be verified *in vitro* and *in vivo*. Third, due to the diverse biological functions of lncRNAs, the lncRNAs that we have not investigated further should not be considered to be unrelated to CHD.

## CONCLUSION

In summary, our study provides a comprehensive profile of AF-derived lncRNAs and mRNAs in VSD foetuses and further reveals that these DE-mRNAs and DE-lncRNAs contribute to the development of VSD *via* cell adhesion, cell proliferation, cell apoptosis, and the Shh signaling pathway. Finally, we constructed a ceRNA network to elucidate the molecular mechanism of lncRNAs in VSD development.

### Funding

This work was supported by the Qihang Fund of Fujian Medical University (grant number 2020QH2039), the Quanzhou City Science & Technology Program of China (grant number 2021C059R), and the Natural Science Foundation of Fujian Province (grant number 2020J01129). The funders had no role in study design, data collection and analysis, decision to publish, or preparation of the manuscript.

### Grant Disclosures

The following grant information was disclosed by the authors:
Qihang Fund of Fujian Medical University: 2020QH2039.

Quanzhou City Science & Technology Program of China: 2021C059R.
Natural Science Foundation of Fujian Province: 2020J01129.

## Competing Interests

The authors declare that they have no competing interests.

## Author Contributions

- Huaming Wang conceived and designed the experiments, performed the experiments, analyzed the data, prepared figures and/or tables, authored or reviewed drafts of the article, and approved the final draft.
- Xi Lin conceived and designed the experiments, prepared figures and/or tables, and approved the final draft.
- Zecheng Wang conceived and designed the experiments, prepared figures and/or tables, and approved the final draft.
- Shaozheng He performed the experiments, prepared figures and/or tables, and approved the final draft.
- Bingtian Dong performed the experiments, authored or reviewed drafts of the article, and approved the final draft.
- Guorong Lyu analyzed the data, authored or reviewed drafts of the article, and approved the final draft.

## Human Ethics

The following information was supplied relating to ethical approvals (*i.e.*, approving body and any reference numbers):

This study was approved by the Medical Ethics Committee of the Second Affiliated Hospital of Fujian Medical University (NO. 2021-073).

## Microarray Data Deposition

The following information was supplied regarding the deposition of microarray data:

The data is available at GEO: GSE204935.

## Data Availability

The raw data is available in the Supplemental File.

## Supplemental Information

Supplemental information for this article can be found online at http://dx.doi.org/10.7717/peerj.14962#supplemental-information.

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
