# Peer review of "Differential lncRNA/mRNA expression profiling and ceRNA network analyses in amniotic fluid from foetuses with ventricular septal defects"

_PeerJ, doi:10.7717/peerj.14962_

## Round 0.1 · original submission · Major Revisions

Reviewers have now commented on your paper; while they believed that your topic is important and interesting, they had a number of significant concerns that prevent the acceptance of your manuscript in its present form. Their critiques are mostly about the experimental design. They indicate a need for additional data or experiments, and all comments require your careful consideration and appropriate response before the manuscript can be re-reviewed and considered for publication.

For your guidance, reviewers' comments are included below.

If you decide to revise the work, please submit a list of changes against each point which is being raised when you submit the revised manuscript.

·

Basic reporting

-Figure 1: Please mention in figure legend what “N” and “T” stands for.

-Figure 1 A – B and C - D: Please label which figure represents lncRNA and mRNA expression.

-Figure 3B and 3C: Please increase resolution of the text in ordinate. It is not easy to read.

-Line 339: Please replace “Disadvantages” with “Limitations”

-Lines 73 and 74: Replace “toetal” with “total” or “foetal”

Experimental design

-In material and method section, authors mentioned that they had 11 samples with VSD and 11 samples for control group (line 101-103). However, only 5 samples for each VSD and control groups were used for microarray analysis (line 189). Why authors preferred to use only 5 VSD and 5 control samples for microarray analysis? And how did authors choose these 5 samples out of 11 samples?

-Line 226: Since ID1 was not verified by qPCR, why authors preferred to include ID1 in CNC network?

Validity of the findings

-There is an inconsistency between figure1 (A,B) and (E,F) and an inconsistency between Figure 1(A,B) and tables S3- S4. Figure 1E&F and table S3 & S4 shows data from samples of N1,N2,N3,N4,N5, and T1-5. However, In Figure 1 A&B, intensity values of samples N2, N3, N4, N5, N7, T2, T3, T4, T5, T6 are shown. Since this data set from same analyze, why authors did not share intensity values of samples N1 and T1, and replaced these with N7 and T6?

- In conclusion sections (Lines 36 and 354), Autor’s mentioned that this study elucidates the molecular mechanism of the epigenetic regulation of lncRNAs in VSD development. What specific epigenetic mechanisms -such as DNA methylation or posttranslational modification of histones- are regulated by lncRNAs that you found? Please discuss your findings about epigenetic regulation of lncRNAs in VSD development in a specific paragraph in discussion section.

Reviewer 2 ·

Basic reporting

Dr. Guorong Lyu published an article (Construction of the amniotic fluid-derived exosomal ceRNA network associated with ventricular septal defect) last year .
Coherent bodies of work was inappropriately subdivided merely to increase publication count.

Experimental design

no comment

Validity of the findings

This study was redundant and derivative of existing work of the aticle (Construction of the amniotic fluid-derived exosomal ceRNA network associated with ventricular septal defect).
Discussion wasn't well stated.

Additional comments

This paper wasn't novel enough.

Reviewer 3 ·

Basic reporting

1. A period needs to be added at the end of all table and figure titles, such as Table 1, Table 2, and Figure 3.

Experimental design

2. The result of this paper is only a description and listing of each figure or table but does not give a good summary, such as the “Differentially expressed lncRNAs and mRNAs” part.
3. The percentage and number of different classes of DE-lncRNAs should be shown in Figure 2A, simultaneously. Also, how are DE-mRNAs classified? More importantly, it would be more meaningful to display the class distribution of the up- and down-regulated DE-mRNAs and DE-lncRNAs.
4. It would be better to complement the expression level distribution of the up- and down-regulated DE-mRNAs and DE-lncRNAs.
5. In Figure B and Figure C, what is the expression distribution of the up-regulated or down-regulated DE-mRNAs and DE-lncRNAs on each chromosome, and is there a tendency?
6. Why the distribution of expression levels of lncRNAs and mRNAs among the normal groups and test groups are very similar? How does the intensity value are normalized? It should show the distribution of unnormalized values to clearly distinguish the difference in lncRNA or mRNA expression between the test group and the normal group.
7. In “Construction of the CNC network and ceRNA network” part, it would be better to show the up-regulated and down-regulated DE-mRNAs and DE-lncRNAs specific CNC and ceRNA network, respectively. Why do you select the four core mRNAs to construct the CNC and ceRNA network?

Validity of the findings

8. It is not accurate to conclude that “Our study elucidates the molecular mechanism of the epigenetic regulation of lncRNAs in VSD” because this paper did not find the epigenetic regulation of lncRNAs.

Additional comments

9. It is not accurate to describe that “Bioinformatics analyses were further used to identify the functional enrichment and signaling pathways of important lncRNAs and mRNAs.” in L24-25 because this paper did not perform the GO biological process and KEGG pathway enrichment analysis by using DE-lncRNAs. The same error description can be found in L28-30.
10. Please cite the corresponding reference of the software and methods and indicate their version number, such as “Agilent Feature Extraction software” in L136, “GeneSpring GX v12.1” in L137, “R-based limma software package” in L140, “R package” in L141, “Gene Ontology (GO)” and “Kyoto Encyclopedia of Genes and Genomes (KEGG)” in L147-148, “miRcode” and “TargetScan” in L161-162, and “Cytoscape” in L166, “SPSS 21.0” in L183.
11. Please cite the reference of the ellipse software and indicate its version in the legend of Figure 4.
12. Please explain how to calculate the Pearson correlation coefficient and what method this paper used in L157.
13. Please unify the fold-change threshold described in L143 (fold-change >2.0) and L194 (fold change ≥ 2).
14. Please indicate the name and version of the R package used for heatmap and volcano plots in L141.
15. Please describe the threshold of the P value and FDR for identifying the GO term and KEGG pathway in L153.
16. Please unify the description of the unit’s name in the full text, such as “10 minutes” in L115 and “30 min” in L128.

---

## Round 0.2 · Minor Revisions

Your manuscript will be ready for publication when the clarifications and minor revisions are made. Editor agrees with the 4th review’s concern and we expect authors to revise the manuscript accordingly.

Best Regards

Reviewer 3 ·

Basic reporting

no comment

Experimental design

no comment

Validity of the findings

no comment

Additional comments

no comment

Reviewer 4 ·

Basic reporting

The manuscript titled Differential lncRNA/mRNA expression profiling and 1 ceRNA network analyses in amniotic fluid from foetuses with ventricular septal defects, by the authors Wang et al. is an interesting study. The experimental design is good, the manuscript reads well, I have few comments that requires clarification.

Experimental design

The materials and methods section suggest 22 pregnant women were included in the study, in two groups of 11 each, however, only 5 patients from each group were included in the results. How were the patients selected.
Second, why was the control group was selected patients with high risk of Down.

Validity of the findings

Please clarify, lines 210-212: Could you please provide the statistical analysis to show the up- and down-regulated lncRNAs were mainly distributed on chromosomes 1 and 2, thank you.

Additional comments

None

---

## Round 0.3 · accepted · Accept

Please ensure you address the suggestion made by the reviewer during the proof stage.

Thank you

Reviewer 4 ·

Basic reporting

Thank you for answering all the queries and resubmitting the manuscript. I have one comment mentioned below.

Experimental design

Thank you for clarifying the experimental design.

Validity of the findings

Looks great, thank you.

Additional comments

Comment: Supplementary table S5 is not mentioned in the body of the manuscript. Please include it in the appropriate section, thank you.